Diagnosing the dangerous demography of manta rays using life history theory

Dulvy Nicholas K. 1 dulvy@sfu.ca
Pardo Sebastián A. 1
Simpfendorfer Colin A. 2
Carlson John K. 3
1 Earth to Ocean Research Group, Department of Biological Sciences, Simon Fraser University , Burnaby, British Columbia , Canada
2 Centre for Sustainable Tropical Fisheries and Aquaculture & School of Earth and Environmental Sciences, James Cook University , Townsville , Australia
3 NOAA/National Marine Fisheries Service, Southeast Fisheries Science Center , Panama City, FL , USA
Bruno John
Electronic publication date: 2014 May 27
Publication date: 2014
Volume: 2
Electronic Location ID: e400
Received 2013 Dec 20; Accepted 2014 May 7
Copyright: © 2014 Dulvy et al.
Copyright year: 2014
Copyright holder: Dulvy et al.
License: This is an open access article, free of all copyright, made available under the Creative Commons Public Domain Dedication. This work may be freely reproduced, distributed, transmitted, modified, built upon, or otherwise used by anyone for any lawful purpose.
License URL: https://creativecommons.org/publicdomain/zero/1.0/

Keywords: CITES, Data-poor fisheries, Life history invariant, Wildlife trade, Euler–Lotka, Population growth rate, Accounting for uncertainty, Von Bertalanffy growth function, Ocean ivory, Chinese medicine

Funding: Natural Science and Engineering Research Council, Canada Canada Research Chairs program Save Our Seas Foundation project #235 US State Department Contribution to IUCN We thank the Natural Science and Engineering Research Council, Canada (NKD, SAP), the Canada Research Chairs program (NKD), Save Our Seas Foundation project #235 (NKD) and the US State Department contribution to IUCN (NKD) for funding. The funders had no role in study design, data collection and analysis, decision to publish, or preparation of the manuscript. Opinions expressed herein are of the authors only and do not imply endorsement by any agency or institution associated with the authors.

==============================
Background. The directed harvest and global trade in the gill plates of mantas, and devil rays, has led to increased fishing pressure and steep population declines in some locations. The slow life history, particularly of the manta rays, is cited as a key reason why such species have little capacity to withstand directed fisheries. Here, we place their life history and demography within the context of other sharks and rays.

Methods. Despite the limited availability of data, we use life history theory and comparative analysis to estimate the intrinsic risk of extinction (as indexed by the maximum intrinsic rate of population increase rmax) for a typical generic manta ray using a variant of the classic Euler–Lotka demographic model. This model requires only three traits to calculate the maximum intrinsic population growth rate rmax: von Bertalanffy growth rate, annual pup production and age at maturity. To account for the uncertainty in life history parameters, we created plausible parameter ranges and propagate these uncertainties through the model to calculate a distribution of the plausible range of rmax values.

Results. The maximum population growth rate rmax of manta ray is most sensitive to the length of the reproductive cycle, and the median rmax of 0.116 year−1 95th percentile [0.089–0.139] is one of the lowest known of the 106 sharks and rays for which we have comparable demographic information.

Discussion. In common with other unprotected, unmanaged, high-value large-bodied sharks and rays the combination of very low population growth rates of manta rays, combined with the high value of their gill rakers and the international nature of trade, is highly likely to lead to rapid depletion and potential local extinction unless a rapid conservation management response occurs worldwide. Furthermore, we show that it is possible to derive important insights into the demography extinction risk of data-poor species using well-established life history theory.

Introduction

The rapid rise in demand for plant and animal products that are traded through international networks has globalised the reach of economically-powerful consumers causing unsustainable depletion of biological resources (Berkes et al., 2006; Lenzen et al., 2012). While we have long understood the challenges of poaching for the illegal ivory trade (Phillis et al., 2013), we are only now just beginning to reveal the enormous scale of trade in aquatic organisms, such as for the live food fish trade (Sadovy & Vincent, 2002), and the dried product trade in shark fins (Clarke et al., 2006), seahorses (Foster & Vincent, 2004), sea cucumbers (Anderson et al., 2011), and fish swim bladders (Clarke, 2004; Sadovy & Cheung, 2003).

A recent emerging international trade in manta and devil ray gill plates is driving overexploitation elevating their extinction risk (IUCN/TRAFFIC, 2013). The high value of gill plates and the international nature of the trade are driving roving bandit dynamics, incentivising serial depletion and a globalized tragedy of the commons (Berkes et al., 2006). If the population growth rate of manta rays is low, this pattern of exploitation could lead to rapid depletions and local extinction of manta populations. There are two described species of manta ray: Manta birostris (Walbaum, 1792), and M. alfredi (Krefft, 1868). These mantas, and at least some of the nine devil rays and at least some of the nine devil rays (Mobula spp.), are reported in national catch statistics and appear in international trade (CITES, 2007; Couturier et al., 2012; Ward-Paige, Davis & Worm, 2013). Manta and devil rays are taken in targeted fisheries and also as a valuable retained bycatch in China, Ghana, India, Indonesia, Mexico, Peru, Philippines, Sri Lanka and Thailand (Couturier et al., 2012; IUCN/TRAFFIC, 2013). Over the past decade the landings of manta and devil rays have risen more than ten-fold from less than 200 tonnes (t) per year in 1998 to a peak of over 5,000 t in 2009 (Ward-Paige, Davis & Worm, 2013). Manta and devil rays are captured for their gill plates and a single mature animal can yield up to 7 kg of gillrakers which can be worth as much as $680 per kg in Chinese markets (Heinrichs et al., 2011; IUCN/TRAFFIC, 2013). Much of the international trade goes to southern China, and to Chinese communities in other countries (Couturier et al., 2012; Heinrichs et al., 2011). For example, one of the authors has seen devil ray gill plates for sale for $396.80 per kg (under the incorrect taxonomic name Dasyatis centroura) in Vancouver, in 2013 (Fig. 1). The trade is currently difficult to monitor because of a lack of international trade codes and species-specific catch and landings data. Despite this, ∼21,000 kg of dried Manta spp. gill plates are traded annually, derived from an estimated >4,500 individual manta rays, and worth US $5 million (Heinrichs et al., 2011; O’Malley, Lee-Brooks & Medd, 2013).

Figure 1 Image of devil ray gill plates for sale in Vancouver.

Gill plates, tentatively identified as from the sickle-fin devil ray Mobula tarapacana (Philippi, 1892), for public sale in downtown Vancouver, British Columbia, Canada on 26th April 2013: photo credit Nicholas K. Dulvy.

Many (46%) of chondrichthyans are Data Deficient (Dulvy et al., 2014), and in comparison to the data-sufficient species we know little of the life history of manta rays Manta birostris, and M. alfredi. This is particularly problematic when their viability is threatened by rapidly emerging fisheries driven by international trade demand, and CITES Non-Detriment Findings are required for continued international trade (Clarke, 2004; Couturier et al., 2012). Both manta rays were listed as Vulnerable on the International Union for the Conservation of Nature Red List of Threatened Species in 2011 because of the inferred global decline due to directed gill-plate fisheries and their inferred slow life histories (Marshall et al., 2011a; Marshall et al., 2011b). Moreover, recognizing this threat, Brazil, Colombia and Ecuador successfully proposed Manta spp. for inclusion in Appendix II of the Convention on International Trade in Endangered Species of Wild Fauna and Flora (CITES). These listings will come into force on 14th September 2014, by which time their international trade will only be allowed if: (1) specimens were legally sourced, and (2) the export is not detrimental to wild populations of the species (a non-detriment finding, NDF) (Mundy-Taylor & Crook, 2013; Vincent et al., 2013). Non-detriment findings rely on the ability to assess the sustainability of removals of individuals for the international trade from national populations. One of the principal challenges of assessing sustainability is that there is often a high degree of uncertainty in the population biology of species, and the pattern and rate of exploitation (Ludwig, Hilborn & Walters, 1993). However, decisions on the sustainability of fisheries and trade often have to be made without the benefit of sufficient information. Recent advances have made it possible to account for sources of uncertainty and this is increasingly an important part of the decision-making process in fisheries management and conservation (Baker & Clapham, 2004; Magnusson, Punt & Hilborn, 2013; Peterman, 2004).

One approach to dealing with uncertainty in life histories is to draw upon life history tradeoff rules that constrain the range of plausible trait values (Beverton & Holt, 1959; Law, 1979). There are fundamental constraints to the acquisition, allocation and metabolism of energy resulting in a narrow set of rules of life (Dulvy & Forrest, 2010; Jennings & Dulvy, 2008). These rules can be used to choose a plausible range of life history traits, which when combined with simple methods to propagate the uncertainty in the true trait value, can be used to provide powerful insights into demography and fisheries sustainability (Beddington & Kirkwood, 2005). Recent work using a simple life history model suggests manta rays are intrinsically sensitive and have low capacity to rebound from even low levels of fishing mortality (Ward-Paige, Davis & Worm, 2013).

Here, we examine the potential risk to manta ray populations from fishing to supply the dried gill plate trade. Specifically, we calculate the maximum intrinsic rate of population increase (rmax) of manta rays, and compare their demography to other sharks and rays. Our model and approach provides a demographic basis for evaluating the sustainability, or otherwise, of manta fisheries in the face of considerable uncertainty in their life history.

Materials and Methods

We first outline the Euler–Lotka life history model and the three key parameters required to estimate the maximum rate of population increase (rmax): the annual rate of production of female offspring α˜, age at maturity (αmat), and the instantaneous natural mortality rate (M). Second, we describe plausible ranges for each of those parameters for a generic manta ray life history. Third, we use a Monte Carlo procedure to propagate the uncertainty these three life history parameters through the Euler–Lotka model to calculate a distribution of the plausible range of manta ray maximum rate of population increase rmax. Finally, we compare the demography of the manta ray to the life histories and demography of 106 other sharks and rays.

We chose to estimate the extinction risk of manta rays by calculating the maximum rate of population increase using a variant of the Euler–Lotka model (García, Lucifora & Myers, 2008; Hutchings et al., 2012). This is one of the oldest and simplest life history models and is founded on the principle that a breeding female only has to produce one mature female in her lifetime to ensure a stable population size (Charnov & Schaffer, 1973; Charnov & Zuo, 2011; Myers & Mertz, 1998; Simpfendorfer, 2005): α˜=ermaxαmat−permaxαmat−1

where α˜ is the annual rate of production of female offspring. Here we calculated α˜ as l/i∗0.5, where is l litter size and i is breeding interval, corrected for sex ratio i.e., 0.5). αmat is age at maturity, and p is the adult survival rate, where p = e−M, where M is the instantaneous natural annual mortality rate yr−1. While local aggregations of manta rays may be sex-biased we assume an even sex ratio at the, wider, species level. The simple elegance of the model is that it requires only estimates of three biological parameters: annual reproductive output α˜, age at maturity (αmat), and natural mortality (M). Two of these parameters are highly uncertain α˜ and αmat) and the other (M) is estimated indirectly, which can also result in uncertainty. Hence, we aim to estimate a range of rmax to encompass the widest range of life histories that are plausible for manta rays and hence would encompass the true parameter values.

The existence of more than one species of manta ray was only recently recognized (Marshall, Compagno & Bennett, 2009); furthermore, with the geographic overlap and in the absence of sufficient evidence to differentiate the life history traits required by the model we thought it most defensible to evaluate a generic manta ray life history.

Annual reproductive output α˜

One pup is produced per litter (rarely two) and gestation period is approximately one year (366–374 days in the Okinawa aquarium) (Couturier et al., 2012). This suggests at least an annual breeding interval, but there may also be a chance of skipped breeding or multiannual reproductive cycles. There is evidence for a biennial cycle where 1 pup is produced every two years (Couturier et al., 2012; Marshall & Bennett, 2010). An even more extreme example is the recent discovery of a complete absence of pregnant females for four years in the Maldive Islands, following three biennial cycles, which could be interpreted as one pup every five years (G Stevens, pers. comm., 2013). Similar patterns of skipped reproduction have been noted in Japanese waters (T Kashiwagi, pers. comm., 2013). It is worth noting that extended periods of ‘non-pregnancy’ may be an artifact of occasional sightings and/or poor viewing angles (Bennett, 2014). As is typical in demographic modeling we assume an even sex ratio. Under these assumptions a plausible range would be an annual reproductive output averaging 0.25–0.5 female pups per year, but we considered extremes out to an annual reproductive output 0.1 (1 female pup every five years). Because of the simple tractable nature of our modeling approach, we did not have the opportunity to consider juvenile mortality. However, juvenile survival may not be of overriding importance for overall population growth rate, because they are likely to have low mortality and contribute relatively little to population growth rate compared to sub-adults (Heppell, Crowder & Menzel, 1999). We expect manta pups to have low mortality due to their extremely large size in comparison to other sharks and rays. Mortality patterns are strongly size-dependent in the ocean and hence larger individuals are likely to have much higher survival rates (Charnov, Gislason & Pope, 2012; Gislason et al., 2010; Pope, Shepherd & Webb, 1994). Manta offspring are some of the largest offspring of any ectotherm in the ocean. The size of birth of manta pups is 130–150 cm disc width, considering the maximum linear dimension this is one of the largest of any elasmobranch. The maximum linear dimensions of offspring sizes of 274 elasmobranchs ranged from 6.8 cm in Cuban pygmy skate (Fenestraja cubensis) to 175 cm in the basking shark (Cetorhinus maximus), and the size at birth disc width of a manta ray of 130–150 cm lies in the upper 95th percentile of the distribution of maximum linear dimension of size at birth or hatch of these elasmobranchs (Cortés, 2000; Goodwin, Dulvy & Reynolds, 2002; Jennings et al., 2008). As survival information becomes available, future models that account for age and stage-specific mortality are likely to provide more nuanced insights into manta ray demography.

Age at maturity (αmat)

Male reef manta rays (M. alfredi) mature at 3–6 years in Hawaii and female maturity is subject to considerable debate, and for our purposes is inferred to be 8–10 years (Marshall et al., 2011b).

Natural mortality (M)

Can be estimated indirectly from the von Bertalanffy growth coefficient (k) or can be assumed to be the reciprocal of lifespan, 1/maximum age (Charnov, Gislason & Pope, 2012; Dulvy et al., 2004; Pauly, 2002). Here we draw inferences from both approaches.

There is no growth curve available for manta rays, however we can draw some inference as to the plausible range because fish growth parameters are narrowly constrained and highly correlated because of fundamental life history tradeoffs (Charnov, Gislason & Pope, 2012). The rate of somatic growth (as indexed by the von Bertalanffy growth coefficient, k) is negatively-related to the asymptotic maximum size (L∞) within a narrow range (Jensen, 1996). Hence, we review the von Bertalanffy growth curves of larger tropical batoids (>1 m) to guide the choice of a plausible range of k for manta rays. The available growth rates for species with similar lifestyles, tropical and subtropical myliobatoid rays (Table 1) and the tropical planktivorous whale shark, reveals that most k values lie between 0.009 yr−1 and 0.28 yr−1 (Table 1). It might be expected manta rays would have k values toward the lower end of this range because they reach a considerably larger size than most of these myliobatoid rays. While known from temperate regions, they are typically found in warm tropical and subtropical water. They are planktivores and hence can access a much larger food resource base and higher growth might be expected at high temperatures. There is some evidence that planktivores grow quickly because their feeding mode is more energetically profitable when individuals (and their gape) reach a larger size. Comparisons to whale shark and the slower growing myliobatoid rays would suggest manta k values around 0.03–0.04 yr−1 (Wintner, 2000).

Table 1 Growth estimates for tropical rays and whale shark.

von Bertalanffy growth parameter estimates for species with similar life styles to the manta rays; (a) tropical myliobatoid rays larger than 1 m total disc width, and (b) the tropical planktivorous whale shark.

	Species
name	IUCN
statusa	Sex	Maximum
length (cm)b	Maximum
age (years)	L ∞	k	Reference	
a.	Mobula japanica	NT	both	310	14	NA	0.28	(Cuevas-Zimbrón et al., 2012)	
	Myliobatis californicus	LC	M	158.7	6	199.1	0.0596	(Martin & Cailliet, 1988)	
	Myliobatis californicus	LC	F	158.7	24	158.7	0.0095	(Martin & Cailliet, 1988)	
	Myliobatis californicus	LC	F	158.7	24	156.6	0.099	(Martin & Cailliet, 1988)	
	Aetobatus flagellum	EN	F	150	19	152.7	0.111	(Yamaguchi, Kawahara & Ito, 2005)	
	Aetobatus flagellum	EN	M	100	9	131.8	0.133	(Yamaguchi, Kawahara & Ito, 2005)	
	Rhinoptera bonasus	NT	both	102	18	123.8	0.075	(Neer & Thompson, 2005)	
b.	Rhincodon typus	VU	NA	1370	NA	1400	0.026–0.051	(García, Lucifora & Myers, 2008; Pauly, 2002)	
Notes.

a IUCN Red List Categories

CR Critically Endangered

EN Endangered

VU Vulnerable

NT Near Threatened

LC Least Concern

DD Data Deficient

b Disc width (cm) for rays and total length (cm) for whale shark.

The maximum age of manta rays can be inferred from the longest period of resightings of individuals through photo identification projects (Town, Marshall & Sethasathien, 2013). In Hawaiʻi one female has been continuously resighted since 1979, providing a minimum estimate of longevity of the Manta alfredi of 31 years (Clark, 2010). The inferred manta ray maximum age of >31 years is considerably higher than the 19 to 25 years for Aetobatis flagellum, Myliobatis californicus and Rhinoptera bonasus, so a more plausible range for k might be 0.05–0.1 yr−1. Life history invariants can be used to estimate mortality from growth rate, assuming an M/k ratio of 0.4 which is more typical for elasmobranch fishes than the higher ratio of M/k = 1.5 observed in teleost fishes and reptiles (Frisk, Miller & Fogarty, 2001). For a range of k of 0.03–0.1, then M is between 0.012 and 0.04 yr−1.

Estimation of maximum intrinsic population growth rate

We model parameters encompassing the following ranges: k = 0.03–0.1, M = 0.012 to 0.04, age at maturity = 8–10 years and an annual reproductive rate of 0.25 to 0.5 female pups per year. To propagate the uncertainty inherent in these parameter ranges, we drew 10,000 values of each parameter from a random uniform distribution bounded by the plausible range of each. While life history traits are typically distributed around a mean value in a Gaussian manner, we choose a more conservative uniform distribution to explore the full range of parameter space. Maximum intrinsic population growth rate was calculated for the 10,000 triplets of α˜, αmat and M by iteratively solving for rmax using the nlminb optimization function in R statistical software version 2.15 (R Core Team, 2013).

Manta rmax compared to other sharks and rays

We compared the manta ray rmax to all available estimates (n = 106), comprising 105 published estimates for chondrichthyans (García, Lucifora & Myers, 2008), to which we added the filter-feeding CITES-listed basking shark (Cetorhinus maximus) which has an M of 0.024 (based on a growth coefficient k of 0.067), age at maturity of 10, and an annual reproductive output of 1.5 females per litter every two years (assuming an 18 month pregnancy) (Pauly, 2002). For plotting, we extracted all maximum sizes as the total length in centimeters, except for Myliobatiformes and Chimaeriformes for which we used disc width and fork length, respectively (García, Lucifora & Myers, 2008; Pauly, 2002). There is wide geographic variation in maximum disc width and many M. alfredi individuals average around 400 cm increasing to 490 cm DW cm (Marshall, Dudgeon & Bennett, 2011). The giant manta ray consistently reaches a maximum size of over 700 cm DW with anecdotal reports of up to 910 cm DW (Marshall, Compagno & Bennett, 2009). Here, for graphical purposes we assumed a maximum size of 600 cm DW.

Results

Assuming that the range of life histories explored encompasses our current knowledge, then the median maximum intrinsic rate of population increase rmax for manta rays is 0.116 (95th percentile = 0.089–0.139, Fig. 2A). The lowest rmax value of 0.079 corresponds to an annual reproductive output, α˜=0.25, αmat = 10 years, and natural mortality, M = 0.04, and the highest rmax of 0.15 corresponding to α˜=0.5, αmat = 8 years, and M = 0.012.

Figure 2 Maximum intrinsic rate of population increase for 106 chondrichthyans, and the manta ray.

(A) Maximum intrinsic rate of population increase for 106 chondrichthyans, including the manta ray. (B) Sensitivity of manta ray maximum intrinsic rate of population increase to variation in natural mortality rate, age at maturity and annual reproductive rate.

The rmax decreases considerably when annual reproductive output is lower. The rmax is most sensitive to annual reproductive output α˜ compared to the age at maturation αmat, note the difference between each α˜ is greater than among growth rates or ages of maturation (Fig. 2B). The sensitivity to annual reproductive output α˜ relative to age at maturation αmat becomes increasingly important when annual reproductive output is low (Fig. 2B). There is a positive relationship between growth (and hence mortality) and rmax across species (Fig. 3A), and larger species have lower rmax (Fig. 3B).

Figure 3 Manta rays have low intrinsic rates of population increase due to their slow growth to a very large size.

Maximum intrinsic rate of population increase versus, (A) von Bertalanffy growth rate k, and (B) maximum linear dimension (cm) for 106 chondrichthyans on a logarithmic scale. Whale and basking sharks are highlighted for comparison. Manta ray mean (black diamond) and 95th percentiles.

Of the 106 species for which we could calculate the maximum intrinsic rate of population increase, the manta ray had one of the lowest rmax values (0.116). The rmax of deepwater sharks (n = 14) is significantly lower than for continental shelf and oceanic pelagic species, as revealed by García, Lucifora & Myers (2008). Aside from the deepwater sharks which are all intrinsically sensitive to overfishing (Simpfendorfer & Kyne, 2009), in shallower water the species with the lowest rmax were the temperate basking shark (Cetorhinus maximus) rmax = 0.109, followed by the manta ray (rmax = 0.116).

We compared the maximum population growth rate rmax as calculated from the modified Euler–Lotka models and the population growth rate r (which equals ln[λ]) as calculated from age-structured models (Cortés, 2002). We found both measure of population growth significantly related, but the slope of the relationship was 0.26 (±0.09 standard error) suggesting rmax is typically four times greater than r (F1,27 = 8.09, p = 0.008, adjusted r2 = 0.2). Hence, in assessing the productivity of species against the criteria of Food and Agriculture Organization of the United Nations (Musick, 1998), it might be more precautionary to estimate r as rmax/4 = 0.029 (95th percentile [0.022–0.35]), and hence manta ray has “very low” productivity (<0.05).

Discussion

We show how life history theory can be used to guide the estimation of an important demographic parameter—the maximum intrinsic rate of population increase rmax—and likely sustainability of even the most difficult-to-study marine animals. Manta rays are data poor but compared to many other chondrichthyans they are still relatively data rich. Of the 1100+ known species, manta rays are among the 106 species for which we can calculate rmax. Nevertheless, the paucity of life history data for manta rays is very typical of the many data-poor fisheries of the world, particularly in the tropics. But the absence of data should not preclude or delay management. Our analysis shows that manta rays have one of the lowest maximum intrinsic rates of population increase of any of the chondrichthyans studied to date. Our approach is designed not to estimate the one true value of the maximum intrinsic population growth rate but to calculate these values while understanding the sensitivity to the input parameters and accounting for uncertainty in those values. Despite some uncertainty in life history traits, the plausible range of manta ray rmax estimates is narrow (Fig. 2), because life history tradeoffs between maximum asymptotic size and the growth rate narrow the parameter space. It is likely that the range is narrower than we show because we could not account for the covariance of life history traits, if we were able to do so this would further narrow the plausible range of manta ray rmax estimates.

We find that the maximum rate of population increase is slightly higher than a recent estimate of the intrinsic rate of population increase, r = 0.042–0.05 (Ward-Paige, Davis & Worm, 2013), compared to our median rmax = 0.116. The range of parameters we used encompassed those of Ward-Paige, Davis & Worm (2013) and suggest the difference in r versus rmax may be due to differences in the method used to estimate natural mortality and that the rebound potential method consistently provides lower population growth rate. We used an elasmobranch-specific mortality estimator (Frisk, Miller & Fogarty, 2001), whereas the other used an estimator based on fishes, molluscs and whales (Hoenig, 1983). A more puzzling issue is why our approach reveals that manta rays have one of the lowest rmax of any chondrichthyan, whereas the other suggests manta rays may have an intermediate r (Ward-Paige, Davis & Worm, 2013). This issue is beyond the scope of this paper, and requires a simulation-based performance comparison of these kinds of models. While close, the difference in demographic estimates underscores the need for a better understanding of such rule-of-thumb mortality estimators and a comparison of the performance of different variants of simple scalar unstructured demographic models, such as the Euler–Lotka model, the rebound potential model , and Pope’s Fjeopardy model (Pope et al., 2000; Simpfendorfer, 2005; Smith, Au & Show, 1998).

Without the opportunity to consider juvenile survival rate, our estimates of rmax may be slightly too high. We implicitly assume that juveniles have the same survival rate as adults. However, a more realistic assumption might be to assume that juvenile survival rate approaches adult survival rate as described by survival to adulthood raised to the power of the age of maturity. Such an approach to juvenile survival would result in smaller rmax values than we present here (EL Charnov, pers. comm., 2013).

One might object to the calculation of rmax given such great uncertainty in basic life history of these data-poor species. However, the pragmatic reality is that we do not have the luxury of waiting for more data to become available. And indeed increasing effort is being paid to understanding safe biological limits for the exploitation of target and bycatch species (Dulvy et al., 2004; Pardo, Cooper & Dulvy, 2012). At the most recent 16th Conference of the Parties of the Convention on the International Trade in Endangered Species both species of manta ray were listed on Appendix II, which includes, “species that are not necessarily now threatened with extinction but that may become so unless trade is closely controlled”. Under this regulation Appendix II species can only be traded subject to three conditions, two of which pertain to the legality of capture and welfare (of live transported species), and the third relates to the sustainability (or otherwise) of trade—the so called non-detriment finding (Vincent et al., 2013). A non-detriment finding confirms that the trade of specimens will not be detrimental to wild populations of the species. A key condition of the CITES listings of both manta rays has been a delay by 18 months until 14th September 2014 (CITES, 2013). By this date, any nation, that is party to the CITES, wishing to trade manta ray gill plates (or other products) needs to develop methods for assessing that proposed trade is sustainable and not detrimental to wild populations. There is very little time in which to gather new data and hence our simple modeling demographic model, constrained by life history tradeoffs and accounting for and propagating biological uncertainty, provides a much-needed first step toward developing methods to support the development of methods to assess the sustainability of exploitation and international trade.

Our analysis reveals that a key parameter to estimate in future field studies are the growth coefficient k from a von Bertalanffy growth curve, fitted appropriately to size-at-age data (Pardo, Cooper & Dulvy, 2013; Smart et al., 2013; Thorson & Simpfendorfer, 2009). Hopefully, the growth coefficient k can be estimated for manta rays, as has been done for other smaller tropical myliobatoids. However, there is a real possibility that annuli may not be recoverable from manta rays because mobulid vertebrate tend to be poorly calcified (WD Smith, pers. comm., 2012). Hence, resighting programmes may be the most pragmatic method of estimating a growth curve (Town, Marshall & Sethasathien, 2013). As we have shown, natural mortality rate depends heavily on k and the ratio of M/k, which is around 0.4 for elasmobranchs (Frisk, Miller & Fogarty, 2001). If it is not possible to estimate a growth curve for manta rays in the near future then demographic modeling will be heavily reliant on our understanding of: (1) the overall pattern of maximum size (L∞) and growth coefficient (k) in elasmobranchs, and especially tropical and subtropical batoids, and (2) the M/k ratio. Future work should concentrate on understanding why the elasmobranch M/k ratio is around 0.4, by comparison the teleost and reptile M/k ratio is around 1.5 (Charnov, Berrigan & Shine, 1993). This ratio has a profound influence on the estimate of population growth rate and the sustainability of species, and hence understanding the life histories, ecological and environmental correlates of the M/k ratio can only improve the predictive power of these simple demographic models.

Other parameters that strongly influence the maximum intrinsic rate of population increase are the age at maturation and the annual reproductive rate. These parameters are very poorly understood (Marshall & Bennett, 2010). The manta ray annual reproductive rate estimates of one pup per year are based on aquarium-held specimens under relatively ideal conditions, and hence these estimates are likely to be optimistic. There is unpublished evidence suggesting that annual reproductive rates may be much, much lower and variable among and within individuals. The proportion of pregnant females returning to long-term (6–8 years) study sites in the Maldives previously suggested a biennial reproductive mode, but in recent years no pregnant females have returned (G Stevens, pers. comm., 2013). The absence of returning pregnant females may indicate a spatial shift of returning females, but also may hint at much lower and more variable annual rates of reproductive output than we have modeled here. We recommend that the demographic rates of manta rays be revised as more details of the temporal and geographic variability in reproductive output come to light. The emerging observations of year-to-year variation in individual reproductive output may lead to variance in year-to-year population growth rate which can only serve to depress the long-term population growth rate further elevating extinction risk (Hutchings, 1999). And indeed such observations caution us to initiate and undertake local analyses of population structure and reproductive activity and to incorporate local variations into local demographic models and assessments contribution to CITES Non-Detriment Findings. Of course the greatest uncertainty, that we have entirely overlooked, is that future demographic estimates would benefit greatly from species-specific estimates of the key life history parameters: growth coefficient k, annual reproductive rate and age at maturity.

Notwithstanding the current uncertainty in the life history of manta rays, given their very low productivity coupled with small localized populations and predictable seasonal aggregations, the unregulated targeting of local Manta populations for their high-value gill plates is unlikely to be sustainable. The largest targeted fisheries and highest mortality occurs in Indonesia, Sri Lanka, India, Peru and Mozambique and these countries have little fisheries monitoring, regulation or effective enforcement. The time to local extinction depends on the size of the population and the rate of fishing mortality. The very low productivity of manta rays mean that even a moderate level of fishing mortality of F = 0.2 (survival = 0.81) would reduce a small population of 100 individuals to fewer than 10 within less than a generation span (11 years). The key challenge this poses is that it leaves little time to mount an effective conservation management response. These serial depletion fisheries are operated by low-income subsistence coastal fishers, often against a backdrop of declining fish stocks. For such fishers the international market demand for valuable Manta and mobulid ray gill plates is likely to provide a desirable income. Such fisheries tend to be unregulated and even if there are protections these are difficult to enforce, which underscores the importance of international trade regulation.

We thank Thomasina Oldfield and Martin Jenkins for motivating this study. This work was originally submitted as a working paper to the Fourth United Nations Food and Agriculture Organization Expert Advisory Panel which met from 3–8th December 2012. We thank María José Juan Jordá and Lucy R. Harrison for constructive comments and Tracy Saxby, Integration and Application Network, University of Maryland Center for Environmental Science (ian.umces.edu/imagelibrary/) for providing the images. Finally, we thank John Bruno, Mike Bennett and Alastair Dove for their insightful comments and critique. Thanks are also due to Eric Charnov who provided unsolicited and welcome comments on the PeerJ Preprint version.

Additional Information and Declarations

Competing Interests

Author Contributions

Data Deposition

The authors declare there are no competing financial interests.

Nicholas K. Dulvy conceived and designed the experiments, analyzed the data, contributed reagents/materials/analysis tools, wrote the paper, prepared figures and/or tables, reviewed drafts of the paper.

Sebastián A. Pardo conceived and designed the experiments, contributed reagents/materials/analysis tools, wrote the paper, reviewed drafts of the paper.

Colin A. Simpfendorfer conceived and designed the experiments, wrote the paper, prepared figures and/or tables, reviewed drafts of the paper.

John K. Carlson conceived and designed the experiments, wrote the paper, reviewed drafts of the paper.

The following information was supplied regarding the deposition of related data:

ChondroRmax140429.xls. Nick Dulvy. figshare.

Retrieved 00:53, Apr 30, 2014 (GMT)

http://dx.doi.org/10.6084/m9.figshare.1009215—See more at: http://figshare.com/preview/_preview/1009215#reserve_citation.

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
