# Peer review of "Diagnosing the dangerous demography of manta rays using life history theory"

_PeerJ, doi:10.7717/peerj.400_

## Round 0.1 · original submission · Minor Revisions

I am very sorry about how long it took to have your paper reviewed.

·

Basic reporting

The paper is a worthy peice of original work that adheres to PeerJ policies. There are a number of minor typos and formatting matters that need be addressed (given in comments to author box). Literature is up-to-date and approriate.
Figures are OK, although one has an error and others could potentially be more informative.

Experimental design

These attributes have been approprately addressed. There is a possiblilty of splitting the two currnely identified manta ray species in the analysis to show that birostris is probably at greater risk.

There is comment and reference to Mobula species in various parts of the MS, but the paper focuses on Manta, these comments about the sister taxon could be removed.

Validity of the findings

Considering the vaguaries of the data for the species in question I consider this to be an acceptable analysis of the data available.

Additional comments

Good to see this, as it provides important information to further the species' conservation. What follows are observations that you may wish to consider in revising the MS.

Specific comments.

Background.
Line 1: Why capitalise “manta” and “manta ray”? This is a non-standard approach; manta is not a proper noun.
Line 3: The term “slow life history” is a curious and technically poor phrase.



Introduction.
Line 64: The sentence starting “Only in the past decade….” seems superfluous, as it is essentially repeated in the following sentence.
Line 72: suggest removal of capitalisation. Devil rays are often taken to include both Manta and Mobula whereas your use suggests otherwise.
Line 74: Authority is “(Walbaum, 1792)”.
Line 80: You should simply state “tonnes (t)” as a tonne is 1000 kg. Although in the USA it is referred to as a ‘metric ton’.
Line 82: “traditional medicine” (is it really traditional, given that trade in gill plates was historically low?).
Line 84: ‘Southern China’ is not a large city as implied.
Line 92: I would suggest that we actually know a fair bit about the biology of manta rays compared to many (most?) other elasmobranch fishes…..
Line 97: You use the plural for manta rays, but the singular for life history, even though the life histories are presumably different.
Line 101: CITES Appendix II listing to occur 14th September 2014.

M&Ms.
Line 148: One interesting thing about manta rays is that they often show biased sex ratios (i.e. ≠ 0.5), although to date the reasons behind these observations are unclear and I suspect that a ratio of close to 1 : 1 actually exists in nature.
Line 158: Manta birostris reaches significantly larger body size (c. 7 m DW) cf. M. alfredi (max. 5.5 m, but commonly 3.5 – 4.5 m DW). The comment about similarity in body size is misleading.
Line 165: Non-annual or biennial reproduction may well occur, although there are inherent problems in determining whether manta rays are pregnant and some of these extended periods of ‘non-pregnancy’ may be an artefact of occasional sightings/poor viewing angle, etc.
Line 174: There are no data on juvenile mortality for manta rays, but it is this period of life that is likely to have the highest mortality rates. I am not sure that this should be disregarded so lightly. Most surveyed manta ray populations are characterised by relatively few (i.e. fewer than expected) small (= young) individuals, suggesting either a failure to recruit into the population (low numbers produced per annum, or high mortality of juveniles) or the juveniles do not join the surveyed populations until they are of a large body size (possible size-based segregation of juveniles).
Line 208: After stating earlier that k might be “on the high end” you then settle for a value towards the low end of the range.
Line 214: Which species was recorded to be at least 31 years of age? This may be important. Earlier, you pointed out that the two manta ray species were of a similar maximum size (even though the difference is about 45%), whereas here with a difference in age of 15 – 30% the value is “considerably higher”.
Line 236: centimetres.
Line 242: Given that there is now a significant difference in size between the two species I was not sure why the analysis could not be done on a species specific basis. This might then show that one species is more ‘at risk’ than the other (although both are highly susceptible to population reductions).
Line 276: “We…”
Line 278: The paucity of data for many other species is much more pronounced. For Manta we have size data, some good age data, some reproductive data (gestation period, reproductive period) and even some (yet to be published) growth estimates. As such maybe Manta is not that data poor.
P314: Not sure why the reference to Mobula occurs here as this paper has no estimates for them.
Line 343: Physical mark-recapture is unlikely for Manta, and photography-based resighting will only be effective if laser photogrammetry (or similar) is used to determine body size ‘accurately’. Manta ray vertebrae are of no value for age estimation, although tail vertebrae in mobulids has potential.
Line 349: Manta birostris probably should not be considered a ‘tropical’ elasmobranch as it is generally found in cold waters (even when within the tropics). Both species appear to dive/forage at depths where the water is more like a temperate environment.
Line 365: Yes, there is much to do. Reproductive failure on a large scale is probably unlikely. Most animals reproduce when they have the opportunity, and for manta rays this might be an annual event, or like some other elasmobranchs they may have a resting year (or two), but longer intervals seem biologically unlikely.

References.
Perhaps “of” rather than “Of” in JFB, and EBF
Marshall et al. 2011. (delete the “c”).
O/Malley et al. Remove word capitalisation. (ditto for Pauly)
Pardo et al 2012. Abbreviations used instead of full citation.
Phillis et al. lacks journal volume/pages.
Town et al. Now published, with journal volume and page info.

Table 1. Perhaps use a different superscript for disc width as it makes maximum length look like square centimetres.
Figure 1: Nice to see Nick has an iPhone (perhaps crop the image?)
Figure 2: (a) Are the other data (species) provided as supplementary information? It would be good to see some of the other species with low rmax identified. (b). The caption does not explain the figure, which has natural mortality and not growth rate. Amat not defined in caption.

·

Basic reporting

“Mantas” or “Manta Rays” used as a common name should not be capitalized unless at the start of a sentence. Mid-sentence it is either italicised "Manta" or “a manta”/”mantas”.

L13 It seems to me that the intrinsic risk of extinction and the maximum intrinsic population growth rate are not the same thing, as implied by the parentheses here.

L65 To support a statement that “Only in the past decade have we begun to reveal the enormous scale of trade in aquatic organisms”, the authors cite a decade old paper. Oops.

L72 Most or all of the paragraph that starts on this line could be deleted. Market price per kg of filter plates is not pertinent to the discussion of life history modeling. I think a simple sentence in the previous paragraph recognizing the role of filter plates as a motivation for harvest would be sufficient context.

L571 Figure 1 is not pertinent to the subject of the paper and should be eliminated.

L584 The readability of Figure 3 is poor. The panels are small and the use of cartoon avatars with arrows to label specific data points makes it hard to understand what the authors are attempting to show us. It would really help their case to adhere to convention and simply use a different point marker for whale shark, basking shark and manta ray.

Experimental design

L163 It would seem to me that a known gestation range that is entirely longer than 1 year (min 377 days) cannot by definition suggest an annual breeding cycle, but rather a 2 year or biennial cycle, no?

L171 The assumption of even sex ratio seems important to the subsequent analyses. Is there any published data to support this? The authors find in their results that ã is the most important parameter influencing rmax and yet this is also the parameter most influenced by sex ratio. I think some additional comment is necessary here.

L191 I thought it was convention for the Von Bertalanffy coefficient to be expressed as a non-italicised capital “K”, not italicised lower case “k”?

L197 I assume “somatic growth rate” k here is the same as the Von Bertalanffy coefficient k from the previous paragraph. Consistent naming of parameters would help the reader

Validity of the findings

No Comments

Additional comments

This is an interesting exploration of the implications of life history traits on intrinsic population growth potential for an important and vulnerable group of marine megafauna, the mobulid rays. In the absence of better empirical data, the authors plug a range of assumed values into a simple model and look at the propagation of variation into the model outputs by using Monte Carlo iterative simulations. For many species, this would probably be an unacceptable approach, but for intractable but threatened megafauna for which we lack some of the most basic data, it’s the best we have and so in that sense the work is valuable starting point. The authors acknowledge this explicitly on line 316, as well they should.

I don’t have major objections, but I do think it is important to stress throughout (perhaps more than they do) that this sort of modelling exploration is a starting point, a preliminary approach to help define better questions rather than to answer them in any substantive fashion. I would accept it with the revisions below and perhaps some condensation of the introduction and discussion sections, which could be tightened up some.

---

## Round 0.2 · accepted · Accept

Thank you authors for your very complete revision and for carefully and calmly responding to the reviewer suggestions. This manuscript was a pleasure to edit with everyone being so professional and courteous. And I think the paper is fantastic. Less than a year ago I was asking around on twitter what we knew about manta population status and demography and the answer was "not much". So this science fills a void for a unique and threatened taxa. Thank you for undertaking this work and for submitting it to PeerJ.